**communications** engineering

# MechRAG: a multimodal large language model for mechanical engineering
Shuang Li ⬤ ✉ & Jonathan Corney

Engineering design and manufacture are inherently multimodal activities in which engineers consult and produce diverse data and representations across various engineering disciplines and product lifecycle stages. Although well-established digital formats exist for these representations, their use remains restricted within specialist applications, creating silos that limit cross-domain integration. Here we introduce mechanical retrieval-augmented generation (MechRAG), a multimodal large language model architecture designed to unify information from multiple engineering representations typically found in computer-aided engineering and computer-aided design environments. Results demonstrate that MechRAG achieves high accuracy in routinely performed mechanical activities such as data-management or classification tasks, and effectively replicates engineer-level reasoning in more inferential and subjective contexts. Our findings suggest that such conversational interfaces enhance engineering productivity, facilitate more interactive paradigms, and drive transformative workflows across various stages of design and manufacturing.

Conversations between professional engineers frequently mix arbitrary amounts of knowledge about general principles and engineering with the specific details of particular projects or companies[1]. For example, an engineer might reasonably ask a human colleague:

1) What are the important design features of a particular component?
2) How many components are made from 4014 steel or require CNC (Computer Numerical Control) machining to be scheduled?
3) How could the cost of this component be reduced?

Answering such questions with current engineering software would need users to specify precisely what data is required and then develop algorithms to be incorporated as plug-ins, or extensions, to traditional computer-aided design (CAD) software operations. For example, questions related to shape features (e.g. machining or assembly features) have motivated research into Automatic Feature Recognition (AFR) systems[2,3]. Similarly, shape classification has led to work on content-based retrieval, where geometric features and dimensions have been computed to provide keys or labels for database searches of 3D CAD archives[4–6].

The advent of LLMs (Large Language Models), which can provide conversational interfaces to large, multimodality datasets, offers the opportunity to create platforms whose responses can be informed by the contents of multiple data silos created by engineering applications (e.g. FEA (Finite Element Analysis) results, CNC part programs) while simultaneously leveraging the general knowledge and technical understanding derived from text documents to address arbitrary questions (i.e. prompts) stated in natural language. This paper introduces MechRAG (Mechanical Retrieval-Augmented Generation), a system designed to address the knowledge gaps in computer-aided engineering (CAE) and CAD data that are currently inaccessible to LLMs.

The successful implementation of the MechRAG architecture depends on enabling LLMs to work effectively with CAE and CAD data. Unlike text, images, and video—whose widely used formats (e.g., PDF, JPEG, MP4) can be easily processed by LLMs using readily available tools that extract semantic content—CAE data such as 3D CAD B-rep models, FEA meshes, metrology point clouds, or XCT voxels, while rich in numerical detail, lacks directly accessible semantic representations. Motivated by this vision the aim of the work reported here is to: create a prompt driven ChatBot that can reason and analyse CAE data relating to specific components made available to it.

To realise this, aim the following objectives were identified:
1) Create a taxonomy of use-cases that define industrial scenarios against which performance can be bench-marked.
2) Extend LLMs with mechanical software and tools. Explore the ways to utilise data acquired from external mechanical software and tools.
3) Develop an LLM-based system capable of understanding input data and answering the listed problems with multi-modality.
4) Experiment with different configurations of LLM-based systems and determine the best performing configuration.
5) Evaluate the proposed system for practical industrial applications.

In this paper, the proposed MechRAG is capable of responding to arbitrary prompts (i.e. questions and tasks) across different genres and levels

School of Engineering, University of Edinburgh, Edinburgh, UK. ✉e-mail: sli63@ed.ac.uk

of difficulty—capabilities that current CAD/CAE systems lack. For multi-modal classification tasks and information queries about individual models, MechRAG achieved high accuracies up to 89.41% and 90.48% respectively. For answering questions and inferences tasks about collection of models/the whole dataset, human engineers report MechRAG achieved human-level performances.

## Related work

This work draws on three bodies of academic work: LLMs, the use of Retrieval Augmentation Generation (RAG) to extend a LLM knowledge and reasoning about CAD/CAM data. The following sections briefly summarize the reported work in these areas:

### LLMs

In the last few years LLMs have made breakthroughs in understanding and generating human language by continuously expanding the size of the model and the size and diversity of the datasets used for training. In the field of Natural Language Processing (NLP), LLMs are widely used for a variety of NLP tasks[7] such as text comprehension, automated question and machine translation, and so on.

Following on the success of LLMs in NLP tasks the generic architecture has been extended to support Vision Language Models (VLMs) and multi-modal LLMs. VLMs have made advancements in multimodal learning by combining visual and linguistic information. These models have demonstrated strong capabilities in tasks such as image description and 'Visual Question Answering' (VQA)[8].

When faced with specific tasks, traditional Deep Neural Networks (DNN) models require supervised learning with task-specific labelled data, often curated over a long time, to learn patterns and make predictions. Fine-tunning of the foundation models with task-specific data is usually required for different downstream tasks. In the contrast, LLMs allows prompting of task descriptions as inputs and seamless adaption for different downstream tasks. Instead of requiring extensive data for each task, LLMs only use a prompt to instruct as to what to do[9]. LLMs understand these instructions because they've been pre-trained on vast amounts of diverse data, including information about many kinds of tasks at various levels[10]. For prompting, different techniques have been developed to better guide LLMs understand and reason, of which the examples are: zero-shot and few-shot prompting and Chain-of-Thought (CoT).

### RAG systems and variants

LLMs rely on the implicit knowledge stored in their parameters when generating text or other modalities of outputs, this reliance constraints them when dealing with knowledge-intensive tasks, especially when it comes to challenges like hallucination, outdated knowledge, and non-transparent, untraceable reasoning processes. RAG has emerged as a promising solution to these issues by incorporating knowledge from external databases[11]. By combining the retrieval and generation components, RAG enables the generation model to dynamically acquire relevant information from external knowledge bases, thus incorporating more accurate and contextually relevant content in the generation process.

Typical RAG follows 3-step process of indexing, retrieval, and generation. *Indexing* starts with the cleaning and extraction of raw data in diverse formats like PDF, HTML and Word documents for textual data, which is then converted into a uniform text format. To accommodate the context limitations of language models, text is segmented into smaller, digestible chunks. These chunks are then embedded using an embedding model and stored in vector stores. The embedded chunks are crucial for enabling similarity searches between queries and relevant indexed chunks in the subsequential phase of retrieval. *Retrieval* first deals with the query and employs the same embedding model to embed the query to a vector representation. It then compares the similarity between the query vector representation and the indexed vectors in the vector store. The vectors are ranked according to the similarity and the top K individuals are selected for augmentation in the subsequential phase. *Generation*. The query and

selected chunks in the original external data are incorporated into the prompt into an LLM. The LLM then may respond to the query with inherent knowledge and the information contained in the provided external knowledge.

RAG has been reported to adapt to various tasks across different fields. For example, Siriwardhana et al.[12] proposed the RAG-end2end model, which makes the RAG technique better adapted to the text generation tasks in the fields of healthcare and news. Ranjit et al.[13] proposed a RAG-based method for medical image report generation. Bran et al. proposed Chem-Crow, a RAG-based agent to adapt and execute standardized synthesis procedures autonomously[14]. RAG has also been applied to enable robots to complete long-horizon tasks in unpredictable settings[15]. Dai et al. employs a dynamic retrieval mechanism to access external databases to enhance the safety and trustworthiness of autonomous driving systems under complex traffic scenarios[16]. RAG's effectiveness in reducing hallucinations for LLMs has been utilised in educational artificial intelligence (AI) tool to help student learning[17]. RAG has been shown to address challenges in Deep Learning for realistic scenarios, such as long-tail visual recognition, where data distributions are skewed, with most samples concentrated in a few classes. Despite its high relevance in realistic productions, classification performance on long-tail distributions still lags behind balanced datasets[18] but has improved with RAG-assisted methods[19].

While most RAG implementations focus on textual information, there is also a wide range of knowledge stored in different structures and modalities, such as images and videos. To this end, multi-modal RAG has been proposed to better incorporate complex formats of information. Some main applications of multi-modal RAG technology focus on two aspects: improving the accuracy of image descriptions[20] and generating more creative and complex images[21,22]. Multi-modal RAG has also been reported to boost the developments on low-code platforms.

### CAE data analysis

Although CAE technologies have been fundamental to the world of engineering and are now highly developed, for tasks such as 3D model generation, modification and simulations, the established software has limited ability to respond to holistic questions about topics like planning and manufacturing where collective and chronological information is required from multiple sources. A few examples of such questions include: 1) what features a component has, or which feature is shared among a collection of components; 2) which group of components are can be considered similar when assessed against particular criteria; 3) ad-hoc comparisons of components using arbitrary measures (e.g. volume, surface areas etc.) Existing solutions to these sorts of question are typically provided by separate software tools that implement approaches ranging from topology analysis[23–25] to the recent application of sub-symbolic AI approaches such as convolutional neural networks (CNNs)[4,5] and graph neural networks (GNNs)[6] for shape classification and recognition, and fine-tuned foundation models[2,3] for AFR.

Despite the gaps mentioned, CAE data has been well utilised to enable and optimise problem-solving in mechanical engineering. In the literature there is no lack of developments on knowledge-based engineering systems using structured and human-readable symbols to mimic the decision-making ability of a human expert in a specific domain, i.e. symbolic AI approaches. In manufacturing, these approaches have been applied in various topics, such as process planning[26,27], machining feature recognition[28,29], tool selection[30,31], and product development[32,33].

## Methods

First, we consider the possible use-cases for an LLM if company specific CAE data was available to respond to prompts, then we outline the architecture and configuration variables of MechRAG and involved data resources.

### Use case for conversational CAE interfaces

The first step in defining an architecture and investigating the configuration parameters of a multi-modal RAG system for CAE is the definition of the

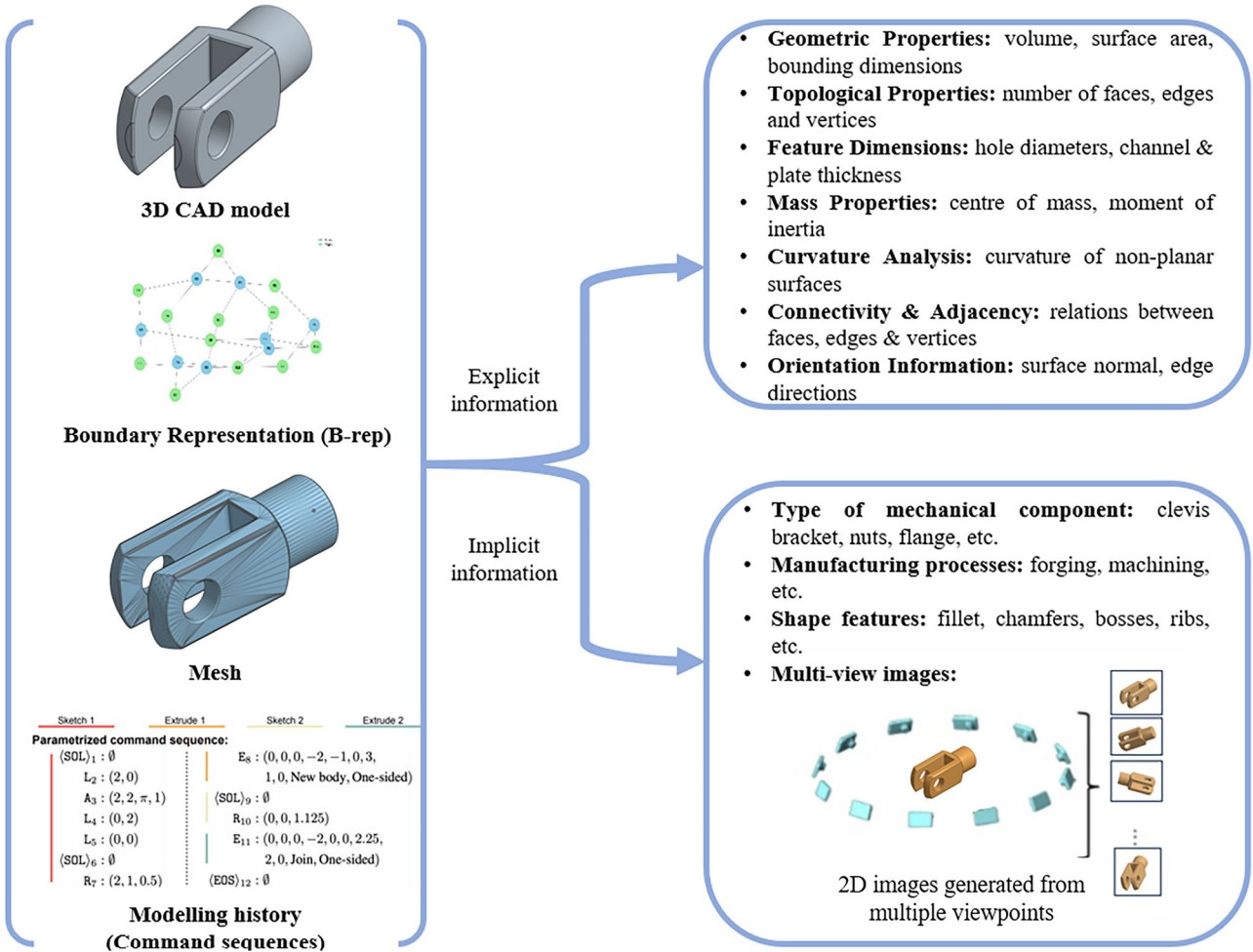

**Fig. 1 | Utilising and transforming computer-aided engineering (CAE) data for large language models (LLMs).** This figure illustrates how original computer-aided engineering (CAE) data—such as CAD models, boundary representations (B-reps), meshes, and modeling histories—can be transformed for use with large language models (LLMs). From these sources, both explicit and implicit information can be extracted, including geometric properties, semantic annotations, and multi-view visual representations, enabling enriched downstream processing and interpretation by LLMs.

system's use-cases. The authors identify that the availability of CAE data to LLM would enable three distinct types of "use case" which are characterises as follows. In all cases it is assumed that all CAE data made available to MechRAG is associated with unique identifying "part number" (e.g., 0021426).

**Level 1 task:.** Direct: request single values for specific parts from CAD or PDM (Product Data Management) data that require no computation. Level 1 functionality provides a uniform, conversational, interface to heterogenous data (e.g., from different CAE packages). Illustrative use case examples include: 1) What is the volume of part 00210008 in cm3? 2) How many round holes does model 00210337 contain? 3) Does part 00210217 have any type of symmetry?

**Level 2 task:.** Collective: request CAE or PDM data for a collection of parts related to the objective of the task Level 2 jobs enable conversations and comparisons among different parts. Illustrative use case examples include: 1) Which parts have the biggest volume and the smallest? 2) What symmetries are present in the dataset of components?

**Level 3 task:.** Emergent: request combined information from both CAD or PDM data, and the general knowledge of LLMs obtained from pre-training. For example concepts of, say, manufacturing processes or relative costs require general knowledge beyond the specifics of a particular CAE dataset. Illustrative use case examples include: 1) Based on part

00217697's modelling history which other components is most similar to it? (The modelling history refers to the command sequences (e.g., sketch and extrusion operations) used to define a component).

### CAE data sources

In this research we have investigated two approaches to bridging the gap between LLMs and the digital representations of CAE data: 1) explicit information from CAE data, such as geometric properties and dimensions, that can be extracted or easily computed (examples shown in Figs. 1), and 2) implicit information of the models, such as the classification of components and view images, that is not directly contained in CAE data, but rather requires resources or software beyond digital representations.

Multiple software can be used to generate multi-view images of CAE data and consequently provide a generic approach to transforming the specialist numerical representations into 2D image data which is interpretable by multi-modal LLMs. For example, Matlab programs can interrogate and manipulate OBJ files, similarly the Solid Edge API provides an interface to STEP files and PyOCC also provides functions to facilitate CAD2Image converter[34] for multi-view image generation from STEP files.

In addition to multi-view images, 3D CAD models also contain numerous metadata which can be extracted and written into textual data. Metadata here refers to geometric features including surface area, model volume, curvature, dimensions, etc. STEP models have the original B-rep information, while STL files typically represent 3D objects by meshes. Complex geometric features that are not well represented in STL files can be

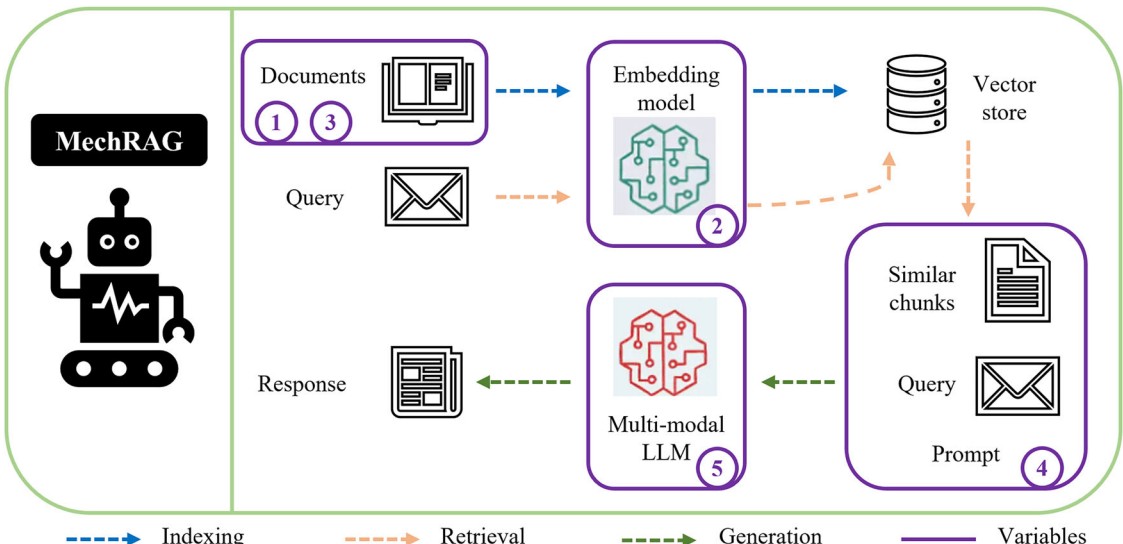

**Fig. 2 | MechRAG architecture.** MechRAG supports multi-modal tasks through a structured architecture comprising three main stages: indexing, retrieval, and generation. The framework explores five configurable variables to optimize performance.

extracted from STEP files, and STL files have better computational utilities available for assessing global properties such as symmetry. In this paper, the python TriMesh package is used to extract geometric data from STL files while PyOCC is used for STEP files. These packages have built-in domain-specific functions to filter and organise big chunks of numerical data in 3D CAD models into features that LLMs can digest.

Another form of information frequently associated with CAE software is their modelling history, effectively sequences of commands (i.e., code) used to construct the model. This general type of representation is investigated here by considering the specific command histories associated with 3D CAD models. The dataset used to enable these investigations is developed in ref. 23. In this dataset, modelling histories consist of 'sketch' and 'extrude' operations. Although simple, the combinations of these two operations are expressive enough to generated a variety collection of 3D shapes[35].

## Datasets

The datasets used for training and assessment of MechRAG comprise of different forms of outputs from mechanical CAE tools and provides an intermediate step from mechanical models in engineering tools to the RAG system.

**Dataset 1 (Mechanical components labelled with their manufacturing process).** Dataset 1 (detailed in ref. 36) contains examples of mechanical parts created by five different manufacturing methods: Fabricated, Forged, Other, Sheet Metal, and Turned. The number of parts for each manufacturing method is about 8,600 and the total number of parts is 43,380. The manufacturing method associated with each component, refers to the main process used to produce the specific part, rather than an exhaustive description of its production process. Parts that are not manufactured by Fabricated, Forged, Sheet Metal, or Turned are labelled as Other. Dataset 1 also contains images of each mechanical components generated from multiple viewpoints.

**Dataset 2 (Mechanical components with shape construction history).** This dataset is derived from DeepCAD dataset[23], which comprises over 170,000 CAD models along with their construction sequences. To meet MechRAG's requirement for both explicit and implicit information, it was necessary to associate Boundary Representation (BRep) models, in STEP format, with the modelling history of each component. This was achieved by identifying the intersection between the DeepCAD and ABC

datasets[37], the latter providing STEP files for CAD models. The resulting dataset includes explicit information and implicit information (in Fig. 1), and modelling histories for 2,131 mechanical components (an example attached in Supplementary Table 2). The modelling histories, originally formatted in JSON, have been translated into natural language descriptions focusing on the operations employed to create the 3D shapes within a CAD system to better enable the conversations with LLMs.

## MechRAG system

Like other RAG systems, MechRAG consists of three stages: indexing, retrieval and generation. However, there are various strategies and alternatives within these three steps, especially for multi-modal RAG. In this paper, we explore different configuration alternatives in different stages of MechRAG. The overall structure of MechRAG and five numbered configuration variables are shown in Fig. 2.

**Embedding method (Variable 1).** The first configuration variable is the feature extraction method for multi-modal data, mixtures of images and text. The feature extraction for text-only data is performed by SentenceTransformer[38]. The first candidate (approach 1) approach for image extraction is multi-modal embedding-based approach: both image data and text data are embedded into a shared vector space using specialized models (such as CLIP[39] (Contrastive Language–Image Pretraining), ViLT[40] (Vision-and-Language Transformer). The goal is to represent different types of data (e.g., text and images) in a unified way, so that they can be compared or used together for retrieval.

An alternative approach (approach 2) is image summary and text embedding-based approach: an image summary (usually in the form of text) is first generated from the image using models like image captioning or summarization models, and then this generated text is embedded into a vector space using a text-only embedding model, i.e., SentenceTransformer.

Compared to approach 2, approach 1 has direct cross-modal alignments and multi-modal understanding, while approach 2 has a more intermediate representation of the images and is text-dominated. Because approach 1 uses both image and text directly, the embedded features are richer representations of the original data, while the image representation quality in approach 2 really depends on how well the image summaries capture the images. However, the strength of approach 2 is that it's less computationally demanding. Once images are summarised, the subsequent tasks no longer deal with images, it is text-only thereafter, which leads to lower requirements on the storage and models processing the embeddings.

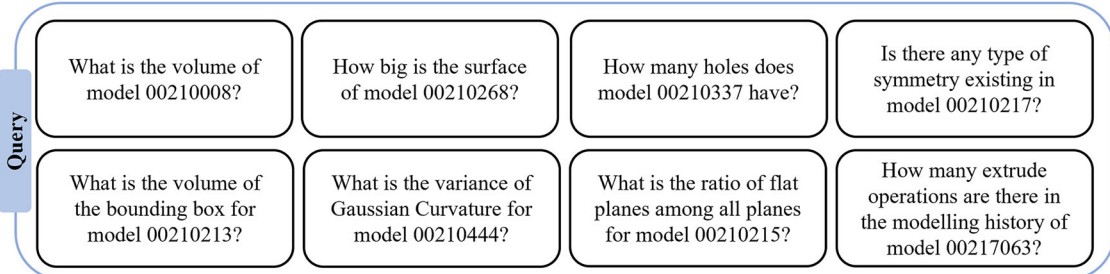

**Fig. 3 | Tasks at Level 1.** This figure presents the set of queries employed for Level 1 tasks across all individual parts.

**Embedding model (Variable 2).** In addition to CLIP, it is also of interest to see how other VLMs perform in terms of feature extractions. This is the second configuration variable considered in this paper, another VLM for embedding, ViLT is also experimentally assessed as an alternative to CLIP. CLIP is a model that jointly trains image and text representations through a contrastive learning approach. ViLT, on the other hand, directly integrates visual and textual information into a unified transformer model[40]. Instead of processing images and text separately, ViLT embeds patches of the image and sequences of the text into a single transformer architecture, which allows it to jointly reason about both modalities in a more integrated manner.

In retrievals, CLIP can take queries of both modalities (text and image) simultaneously while single modality quires, text or image, can also be used to generated embeddings. In contrast, ViLT required both modalities in the same time. In this paper, when single-modal queries are used in ViLT, the input for the other modality is padded empty data, i.e. spaces filled strings for text and white coloured pixels padded maps for images (which is the background colour of the multi-view images).

**Indexing ratio (Variable 3).** Another configuration variable in the indexing phase is the ratio of the dataset used for indexing and therefore accessible for retrievals and prompting. Traditionally, tasks such as classification and recognition are achieved by task-specifically trained supervise learning DNNs, where the train/test ratio is usually big for normal- or small-sized datasets like the datasets here. Common train/test ratios are 9:1 or 8:2, which means most of individuals in the dataset are used for training. LLMs and RAG systems alleviate the demand on specific training for DNNs and big training datasets.

This third configuration variable is set to explore how well LLMs and RAG generalise to tasks that are not targeted in the pre-training of LLMs with much smaller train/text ratios.

In addition to the three configuration variables associated with the indexing phase, there are also freedoms to traverse in the generation phase. Two prompt strategies are compared in this paper, simple prompt and professional prompt, to investigate how prompt engineering affects the performance of the RAG system.

**Prompting strategy (Variable 4).** The simple prompting approach gives a direct task description and desired output format, emphasizing the mechanical engineer's professional background and task objectives. The simple prompt provides the basic analytical framework that guides LLMs in analysing the main manufacturing methods and key features of mechanical parts (illustrated in Supplementary Fig. 1).

The professional prompting approach, on the other hand, employs a more sophisticated strategy that combines, so-called, "few-shot" learning and the CoT technique. In addition to the inputs in the simple prompting, it provides a detailed step-by-step guide to the analysis, including visual inspection, comparison with retrieved similar parts, manufacturing method analysis and key feature identification. This approach is designed to guide the model through a more in-depth, systematic analysis. The strength of the professional prompt strategy is primarily in its structured approach to analysis. By providing detailed analysis steps, professional prompt guides

the model through a more comprehensive and systematic observation and reasoning. This approach not only helps the model to capture more details, but also facilitates a clearer distinction between different categories in classifications (illustrated in Supplementary Fig. 2).

**LLM platform (Variable 5).** The fifth configuration variable is the choice of Multi-modal LLM used for the final generation stage. GPT-4o is an obvious candidate since it has demonstrated capabilities in natural language and multi-modal data processing and generation. However, it is not the only system available, Claude-3.5-Sonnet, another popular multi-modal LLM is also used in the configuration experiments for comparisons.

## Results
In the experiments performed in this section, we assess MechRAG's capabilities and potentials with example tasks ranging from level 1 to 3. LangChain is used in this paper for MechRAG frameworks and ChromaDB is used for vector storage. All experiments are run with Google Colab Pro+ A100 GPUs and up to 89 GB of RAM.

### Experiments for Level 1 tasks
At Level 1 tasks MechRAG is acting as a conversational data management tool with knowledge of all information generated by CAE software for individual parts, and provides access to this dataset via queries in natural languages. The list of queries related to the properties of individual parts is shown in Fig. 3 and were performed to assess MechRAG responses for Level 1 tasks. The specific model identification numbers shown in Fig. 3 are illustrative placeholders within the text of the prompts. In the assessment, the same queries iterated through the component IDs of all the models.

The accuracy of tasks listed in Fig. 3 at Level 1 achieved an accuracy of 90.48%. However, RAG systems are constrained by the context window—the maximum number of tokens a language model can process per query. A consequence of this limit is that when generating a response an LLM might only have access to the retrieved chunks from the whole dataset, rather than the whole dataset. Because of this retrieval process performed here is for individual parts information, there are possibilities where the information for a specific part is incomplete and the accuracy level is not 100%.

### Experiments for Level 2 tasks
For Level 2 tasks MechRAG gives a collective view over multiple mechanical components with related CAE data, enabling comparisons among these components. Example queries of tasks tested for Level 2 are listed in Fig. 4.

The ground truths in Fig. 4 are labelled to compare against the generations. It is noteworthy that for Level 2 tasks, MechRAG is also deeply influenced by the token limits of LLM inputs. The models and the quantity of models exposed to LLMs via augmentations are varying for different generations. Therefore, the ground truths here not only label the correct orders and values, but also scope the present models for generations. Models used to augment MechRAG and generate the results are shown in Fig. 5.

Exposed to the same group of models, multiple experiments are performed and the stability mark (right-hand column) in Fig. 4 details the results of the self-consistency evaluation. To assess response stability, each

**Fig. 4 | Tasks at Level 2.** This figure showcases MechRAG's generated responses for Level 2 tasks. As the number of numeric values increases, the quality and accuracy of the responses tend to decline, highlighting a limitation in handling complex quantitative inputs.

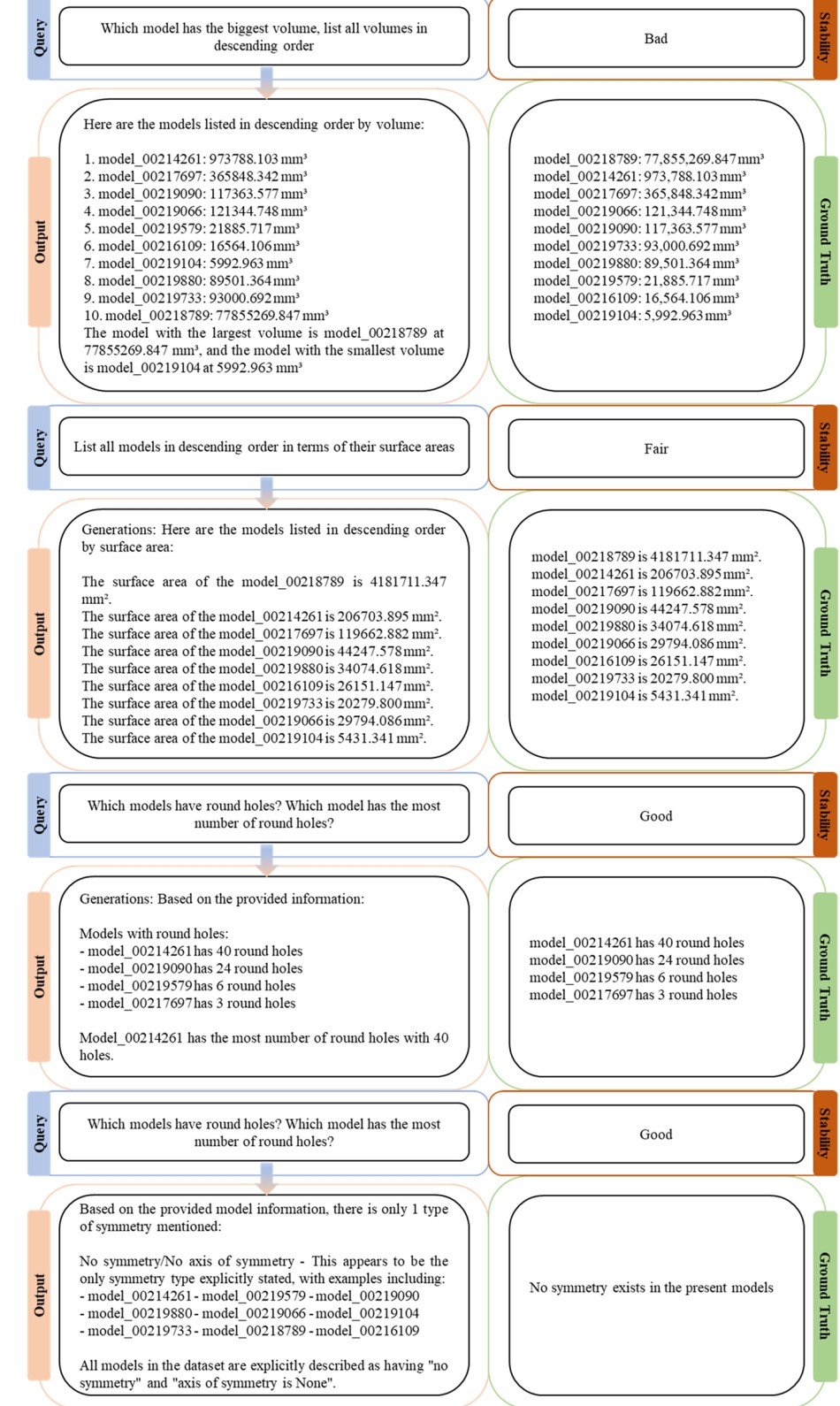

query was repeated 9 times during experiments. Based on the number of correct responses compared to the ground truth, results were categorized as follows: 1) Bad: 1–3 correct responses; 2) Fair: 4–6 correct responses; 3) Good: 7–9 correct responses. It is obvious that when the quantity of numeric values processed increases, the responses tend to become less correct.

Examples of wrong numerical reasonings in Fig. 4 are: the ordering of 117363.577 mm³ 121344.748 mm³ for model volumes, and the ordering of 26151.147 mm², 20279.800 mm² and 29794.086 mm² for model surface areas. These mistakes cause instabilities in consistency of responses and lead to different answers.

**Fig. 5 | Images of models used in generations.** Images of individual parts used for the reported generations.

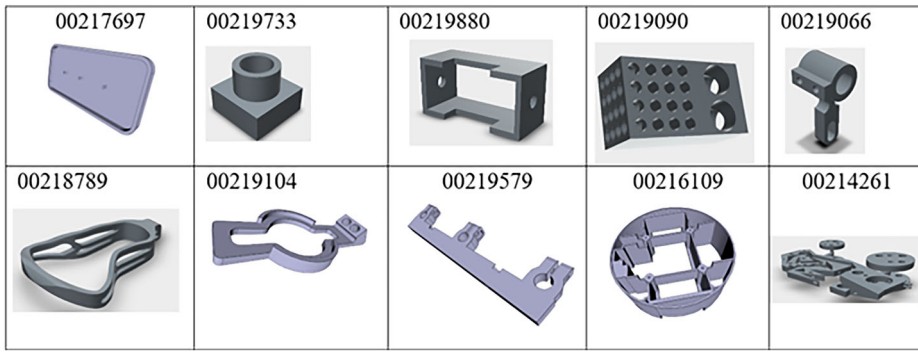

## Experiments for Level 3 tasks

For Level 1 and Level 2 tasks, MechRAG uses CAE data provided at query time to answer questions about mechanical components. At Level 3, MechRAG tackles tasks that require the integration of retrieved, component-specific information (via the RAG process) with the pretrained knowledge of the LLM. In other words, Level 3 prompts elicit responses that combine context-specific CAE data with general production engineering principles (learned during the model's pretraining).

**Conversations with MechRAG.** First textual-data based tasks at Level 3 are investigated and examples are shown in Fig. 6 and Fig. 7. These queries are abstract and have no unique answers, often human engineers would need to debate possible responses to arrive at a consensus view. We evaluated MechRAG with a human study of practising engineers: 106 industrial CPD (continuing professional development) students were invited and 21 volunteered to rate 4 MechRAG response in Fig. 6 and Fig. 7 on a four-point anchored rubric (1 = unacceptable … 4 = fully adequate). The study followed the author's institution's ethics protocol, with informed consent, secure data handling, no compensation, no grading impact, and no supervisory authority from the authors or teaching staff. Our conclusion—that MechRAG's responses were satisfactory and broadly comparable to engineer performance given task ambiguity and limited context—rests on the share of ratings judged "satisfactory" ( ≥ 3/4).While the prompts could be made more precise (for example, by defining what constitutes a singular component or by including batch size or material costs in prompts that seek to identify the most expensive component) they are sufficient to demonstrate MechRAG's core capability: the ability to integrate specific component-level data with general engineering knowledge. Importantly, Level 3 responses are not evaluated on factual correctness alone, but on the appropriate use and synthesis of information sources, including both retrieved CAE data and the LLM's pretrained domain knowledge. It is interesting to note in Fig. 6, that the same model, 00216109, is suggested as both the most singular and expensive to manufacture. The reasoning processes show that although the two prompts concerned different criteria, singularity and expense, similar evaluation metrics, such as number and variety of holes, are used to determine responses to both prompts.

In Fig. 7, first MechRAG is required to identify the most similar models based on the modelling history. The response generation shows a credible process of reasoning about a part's modelling histories. Based on this similarity comparison, a natural next step is to generate modelling histories for models similar to existing ones as shown in the second half of Fig. 7. The queried unknown model has geometric features randomly generated based on model 00219880, and the generation not only identifies its analogue but also suggests modelling history.

**Multi-modal tasks at Level 3.** MechRAG is not limited to textual data, in this section MechRAG is used for identifying the manufacturing methods implicit in the shape of the mechanical components in the scope of

Dataset 1 which contains multi-modal image and text data for the mechanical components.

For this manufacturing method identification task, the queries input into MechRAG are view images of parts instead of text queries as in other tasks. With the image queries, MechRAG generates relevant manufacturing processes with its general knowledge and the augmentation information. To evaluate the generated results, a mapping method is applied to account for the randomness and variety of generated phrases which could essentially refer to the same manufacturing process. (As illustrated in Supplementary Fig. 3, all descriptions under the same category are considered equivalent.)

Because multi-modal RAG and LLMs involve more configuration parameters, first different configuration parameters have been investigated to compare their effects on the performances and obtain the best configuring strategies. All the configuration variables are split into two groups based on the stages of the RAG system they act in. MechRAG system with CLIP as the embedding model for multi-modal data trained on 10% of the whole dataset and prompted using the simple prompting strategy is considered as benchmark for comparisons. The benchmark performance for the manufacturing method identification task is 80.78% accurate.

In the indexing stage, the embedding of the dataset can be implemented in different ways. First two of the most popular embedding models, CLIP and ViLT, have been applied here for multi-model embeddings. In Fig. 8, ViLT is contributing negatively to the RAG system performance compared to the benchmark configuration. In addition to multi-modal embedding, image summary and text embedding has also been explored. Because inevitably a text summary only captures a portion of the information implicitly contained in an image. Since this represents partial knowledge (compared to raw images) incorporating image summaries and text embeddings lead to poorer performance.

Before the advent of LLMs, classifications of image contents were done by task-specifically trained supervise learning DNNs, where the train/test ratio is usually big for normal-sized datasets like Dataset 1. Common train/test ratios are 9:1 or 8:2. LLMs and RAG systems alleviate the demand on specific training for DNNs and big training datasets. Here different train/test ratios are also investigated here to verify exactly how well LLMs and RAG generalise to tasks that are not targeted in the pre-training of LLMs. In Fig. 8, it is clear that MechRAG behaves similar to supervise learning DNNs, where more data is exposed in the training process, the better inference capabilities the model gains. 1%, 2% and 5% of the whole dataset exposed in the training process lead to worse performances compared to the benchmark configuration which has 10% of the whole dataset as training data, while 20% of the whole dataset exposed in training, in the contrast, leads to better performances. However, the significance here is that 10% or 20% of the whole dataset used for training is already providing accuracies of above 80%. This means the proposed MechRAG combining the strengths of LLMs and RAG systems removes the need for large training data and only requires small portions of the whole dataset to provide satisfying performances for classification tasks.

The second group of configuration variables are concerned with the generation stage of RAG systems. From No RAG at all, to simple

**Fig. 6 | Tasks at Level 3 I.** MechRAG responses for Level 3 tasks.

prompting, and CoT, both LLMs, GPT-4o and Claude present increasing better performances. With the help of simple prompting (illustrated in Supplementary Fig. 1), MechRAG reported notably better performance compared to when no information is fed in to generation in Table 1. CoT (illustrated in Supplementary Fig. 2), further adds to the superiority, continues to improve the performances for both LLMs as shown in Table 1. Comparing two LLMs, Claude and GPT-4o, for this manufacturing method classification task, in all three scenarios, no RAG, simple prompting and CoT, GPT-4o is outperforming Claude in terms of all metrics for most of the time.

To conclude, the best performing configuration setting for MechRAG achieves 89.41% accurate on manufacturing process identifications, while inappropriate configurations can set the accuracies back to as low as 38%.

More detailed experiment results of the performances of different MechRAG are shown in Supplementary Table 1.

**Conclusion and discussion**
In this paper, a RAG-based system, MechRAG is proposed to allow CAD/CAM systems to address questions about mechanical components in manners that are currently done by laborious manual processes. MechRAG is capable of questions and tasks across different genres and levels of difficulty. For multi-modal classification tasks and information queries about single models, MechRAG achieved high accuracies up to 89.41% and 90.48% respectively. For answering questions and inferences tasks about collection of models/the whole dataset, human engineers report MechRAG achieved human-level performances.

**Fig. 7 | Tasks at Level 3 II.** More MechRAG responses for Level 3 tasks, concerning the modelling history data.

## Token limits

One clear limitation of the current implementation of MechRAG is the token limit of the context window that can be fed into LLMs in the generation stage, which means the amount of data that can be used to augment LLMs is constrained. With this disadvantage, when dealing with tasks at Level 1 and Level 2, MechRAG can have access to incomplete information regarding the queries and consequently provide inaccurate outcomes.

To illustrate this issue, Fig. 9 shows the effects of token limits of LLMs with Level tasks. In the first scenario, where the whole dataset size equals to the token limit, which means the whole dataset can be input

**Fig. 8 | Performances of different MechRAG configurations.** A summary of performances of different MechRAG configurations. The benchmark setup includes CLIP as the embedding model, a multi-modal embedding strategy, 10% of the full dataset used for training, and simple prompting in GPT. Among the tested variations, only increasing the training data ratio and incorporating the chain-of-thought (CoT) prompting strategy resulted in improved performance over the benchmark.

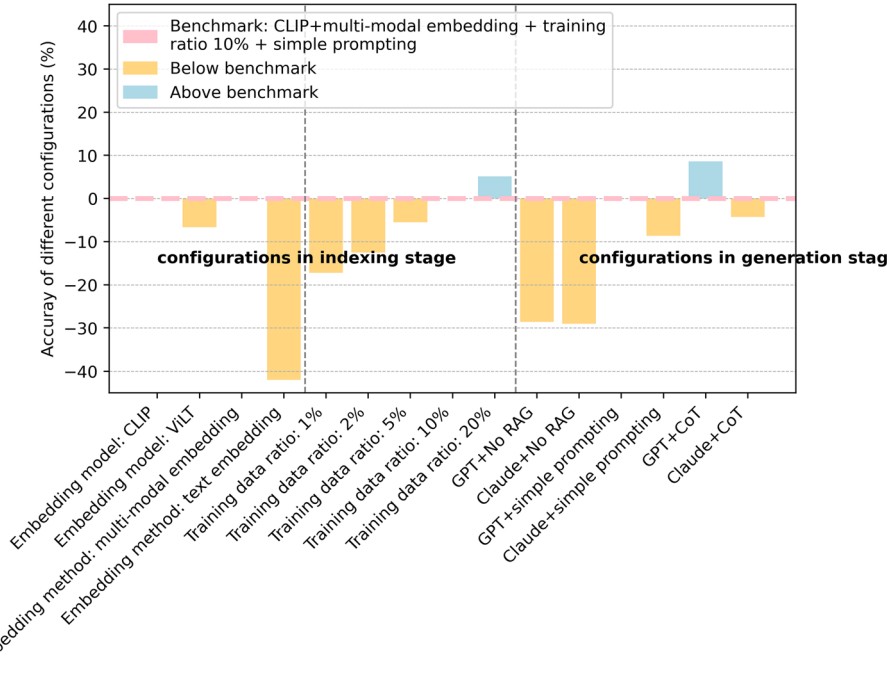

## Table 1 | Performance of different MechRAG configurations

| Configuration | Accuracy | Precision | F1 score |
|---|---|---|---|
| GPT + CLIP | 80.78% | 82.16% | 79.78% |
| GPT+ViLT | 74.12% | 76.07% | 72.93% |
| GPT+image summary and text embedding | 38.82% | 52.74% | 38.25% |
| GPT + 1% data | 63.53% | 65.11% | 62.45% |
| GPT + 2% data | 68.24% | 70.31% | 66.38% |
| GPT + 5% data | 75.29% | 77.21% | 73.31% |
| GPT + 20% data | 85.88% | 86.91% | 85.39% |
| Claude | 72.16% | 77.53% | 70.09% |
| GPT+No RAG | 52.16% | 55.40% | 51.84% |
| Claude+No RAG | 51.76% | 59.01% | 46.49% |
| Claude+CoT | 76.47% | 100.00% | 85.54% |
| GPT+CoT | 89.41% | 100.00% | 94.05% |

This table summarises the performance of different MechRAG configurations based on accuracy, precision, and F1 scores. Among all configurations, the chain-of-thought (CoT) prompting strategy implemented in GPT-4o delivered the highest overall performance.

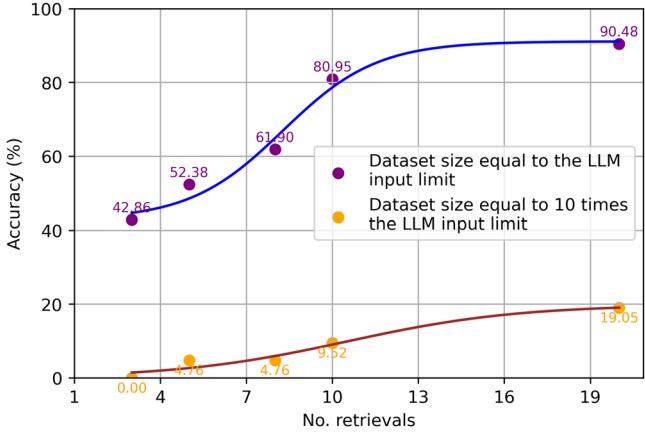

**Fig. 9 | Number of retrievals' effects on the query answering accuracies.** This figure illustrates how the token limit of large language models (LLMs) affects MechRAG's ability to access complete information during generation. In scenarios where the dataset size is within the token limit, performance improves with more retrievals, reaching over 90% accuracy for Level 1 tasks. However, when the dataset exceeds the token limit, additional retrievals yield only marginal gains due to limited access to relevant data.

into LLM and provide accurate information for Level 1 tasks regarding individual parts. When increasing the number of retrievals in this scenario, it is increasing the portion of the whole dataset that is input into LLM. The bigger the portion is, the more likely that the target mechanical part is input into LLM, therefor LLM performance improves. In the top right corner of Fig. 9, approximately the whole dataset is input into LLM, and consequently it is above 90% accurate when querying about Level 1 tasks problems concerning volumes and surface areas of specific models. In the second scenario, where the whole dataset is much bigger than the token limit, using the maximum allowed retrievals is the best shot for retrieving the target models. Decreasing the number of retrievals used only delimits the possibility of LLM accessing the ground truth data, which is well reflected in Fig. 9.

The textual data associated with individual mechanical parts in Dataset 2 ranges from approximately 1500 to 90,000 tokens. Given that the context window of GPT-4o is limited to 128,000 tokens, this suggests that even relatively small assemblies—comprising only a few dozen parts—can exceed the model's capacity for effective context handling. This limitation becomes even more pronounced in the case of complex systems, such as automotive engines or industrial robots, where the number and complexity of components are substantially greater. The resulted performances can be worse than the second scenario in Fig. 9.

## Future work

For future research, there are more mechanical engineering modules to be intergrade into MechRAG system. The datasets used in this paper are collections of discrete mechanical parts, MechRAG will be more impactful if deployed with databases maintained by companies or factories for specific products or production lines. When it's product-specific or production

**Article**

line-specific, more engineering data sources, such as process planning MRP (Material Requirements Planning) data and EBOM (Engineering Bill of Materials) data, can be integrated with CAE data as inputs for MechRAG. This integration provides a more holistic and product-life-cycle scale perspective, and ultimately enable more efficient and optimised decision-making. In addition, MechRAG can be extended to an agentic mechanical engineering AI with real-time, iterative and automatic interactions and feedback loops between users of MechRAG, the conversational interface of MechRAG and the underpinning mechanical engineering software. Such agentic tool can well fit and initiate realistic production tasks such as modifications of shapes or simulations of physical or production processes with queries to MechRAG interface.

Another direction for future research is optimizing the integration of CAE data into LLMs for practical industrial applications. While context window token limits are expected to continue expanding, it remains essential to develop more efficient methods for preparing and representing CAE data that minimize computational resource requirements. As the scope of MechRAG increases its benchmarking against classical approaches to traditional engineering tasks will become possible so, assessment will also become an important part of future research in this area.

## Data availability
Data available in public repositories that issues datasets[23,36]; further inquiries can be directed to the corresponding author.

## Code availability
All code supported this work can be obtained from the following publicly accessible GitHub page: https://github.com/SydneyGit/MechRAG.

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

## Acknowledgements

The authors would like to thank Jason Chi Hao Kwan, Hanyu Wang, and Zhenghao Li, students at University of Edinburgh, for their valuable contributions to this work through data preparation and early-stage concept validation conducted as part of their theses. Their efforts have been instrumental in supporting the foundational aspects of this research. This work was enabled by seed funding from the University of Edinburgh's Generative AI Laboratory (GAIL).

## Author contributions

S.L. contributed to the conceptualisation, implementation and validation of the methodology. The original draft was written by S.L.J.C. contributed to the conceptualisation of the methodology, and the editing of the manuscript. Correspondence to Shuang Li sli63@ed.ac.uk.

## Competing interests

The authors declare no competing interests.
