## [Transparent Peer Review file · Communications Engineering]

MechRAG: A multimodal large language model for Mechanical Engineering

Corresponding Author: Dr Shuang Li

Version 0:

Reviewer comments:

Reviewer #1

(Remarks to the Author)

Dear authors,

thank you for this interesting manuscript for Communications Engineering. The current advance of LLM use in engineering disciplines is manifold and your contribution of a chat-bot for CAD/CAE-based engineering data retrieval and analysis is valuable. There are some potentials to improve the manuscript which I briefly outline below:

- Framing: In the beginning of your manuscript, you make some bold statements such as "create a conversational interface that can respond with a holistic knowledge of all aspects of mechanical engineering". I would suggest to make it a size smaller. Although I understand the long-term objective, the system you describe is from my understanding far away from this. Perhaps it would be good to state it a bit more neutral as I did above in my comment on the chat-bot.

- Literature work in the introduction: Please also use sources for the main statements in your introduction.

- CAE data analysis: I wonder why you limit yourself on sub-symbolic AI approaches. There are plenty knowledge-based engineering systems using symbolic AI that can support or complement your approach and should be at least mentioned in a holistic view on this research field.

- Method description: Please correct the number of use cases. In the body text you mention four, in the list you mention only three.

- Level 1 to 3 tasks: You state queries here. I understand what you mean by this but it would be favorable if you could abstract this to a higher level. E.g. Level one targets at data that is accessible either by CAD-part properties or PDM system outputs for a distinct part number without computation, level 2 involves PDM for multiple part numbers, Level 3 addresses other data structures or, as you state correctly for the experiments part, the answering capabilities of the LLM backbone.

- Data sources: Please explain a bit more in detail what you understand as implicit knowledge. From my point of view, everything you involve is explicit knowledge which can be retrieved e.g. from work plans, PLM systems, etc. You are right when you restrict this simply to the CAD-model.

- Embedding method: In some other papers, the authors made good experience with BLIP2 as data extraction method for multi-modal data. Something that accounts for all your choices: Please explain why you chose the methods/algorithms and what would be alternatives?

- Level 3 Experiments: I'm not convinced of your experiments. Just relying on the examples you state as dialogue, I would not agree that the system is able to give you any information. From my understanding, the question which part is the most expensive in manufacturing there is plenty of information missing to judge this and which are highly context sensitive (lot size, necessity for jigs, process chains involved, precision and tolerances, achievable surface tolerances, ...) which the system is incapable to either retrieve or interpret without further instructions. The same accounts for the history generation. Without a clear purpose of the part, the modeling history is basically interesting but you need to get to a part geometry anyways. So please reflect on the relevance of the level 3 experiments.

- Future work: This sub-section is a bit flat. Although I fully understand the direction, I would recommend to make it a bit more substantial and name concrete steps either for integrating additional data sources (FEM would be interesting but is a completely other data structure and already poses some challenges in CAD-FEA-integration) or in enhancing the result quality of the LLM-based agents.

Reviewer #2

(Remarks to the Author)

This paper introduces MechRAG (Mechanical Retrieval-Augmented Generation), a multimodal large language model architecture designed to unify information from multiple engineering representations typically found in Computer-Aided Engineering (CAE) and Computer-Aided Design (CAD) environments.

The authors posit that MechRAG attains a high degree of accuracy in the execution of mechanical activities that are commonly performed, including data management and classification tasks. They further contend that MechRAG effectively replicates engineer-level reasoning in contexts that are more inferential and subjective. It is suggested that such conversational interfaces have the capacity to enhance engineering productivity to a significant degree, facilitate new interaction paradigms, and drive transformative workflows across a variety of stages in the design and manufacturing process.

MechRAG has the capacity to amalgamate a variety of informational sources, encompassing both visual and textual elements.

The paper sets out a three-tiered structure for tasks, with each tier exhibiting a distinct level of demand. The levels are designated as direct, collective and emergent. The integration of diverse informational sources is facilitated by the utilisation of embedding methods, thereby enabling the execution of varied experiments at different levels. A total of five different configurations of variables are tested for MechRAG, as illustrated in Figure 3. MechRAG's capabilities and potential are evaluated through a series of tasks that span levels 1 to 3. The assessment encompasses a diverse range of question types and levels of difficulty, thereby ensuring a comprehensive evaluation of the candidate's aptitude and proficiency.

In the context of multi-modal classification tasks and information queries concerning individual models, MechRAG demonstrated a noteworthy level of proficiency.

The accuracy levels achieved were 89.41% and 90.48%, respectively. The following text is intended to assist with the answering of questions and the completion of inferences tasks.

In relation to the aggregation of models and the entire dataset, human engineers have reported that MechRAG has achieved a level of proficiency comparable to that of human operators.

The following section will address the topic of performances. A salient constraint of the present implementation of MechRAG pertains to the limited amount of data that can be utilised to augment LLMs.

The work presented is of interest, and the various experiments conducted with different variable configurations are also of interest. The authors employed the system to address repetitive and classification tasks at varying levels of complexity. It is evident that this instrument has the capacity to enhance the productivity of engineers, constituting a significant research endeavour.

Finally, what will be the possible improvements that the authors can imagine for MechRAG in addition of the token limits to improve the performance levels?

Reviewer #3

(Remarks to the Author)

Review of "MechRAG:

#####

A multimodal LLM for Mechanical Engineering" by Shuang Li and Jonathan Corney

The authors introduce a multi-modal large language model (LLM) architecture, Mechanical Retrieval-Augmented Generation (MechRAG). For this, the contribution starts with a motivation, based on this a vision and objectives. Related work is highlighted for the areas of LLMs, Retrieval-Augmented Generation (RAG)-systems, and Computer-Aided Engineering (CAE) data analysis. On this foundation, the method of the work is introduced, including three levels of tasks associated to engineering activities and data, which are addressed by the developed solution, as well as related data sources and datasets. The three stages of MechRAG, indexing, retrieval, and generation, are highlighted and five configuration variables for MechRAG-setups and related experiments are described. The variables are embedding method, embedding model, indexing ratio, prompting strategy, and LLM platform. Then, experiments for the three defined levels are executed and the related results discussed. The contribution ends with a conclusion, discussion and outlook to future work.

Abstract, introduction, and conclusion are clear and appropriate. References contain essential pre-existing work as far as known.

Key result and major claim of the paper is the development and evaluation of a multi-modal LLM architecture for engineering data, and related analysis tasks, respectively. This is a novel approach with a high degree of originality and may be of high interest to the community and potentially beyond.

The contribution is well written and convincing; the chain of reasoning is good to follow and adequately presented. The contribution has the potential to influence the thinking in the area of software support for mechanical engineering data analysis and creation.

Validity of the approach is given based on its high degree of thoughtfulness and structure. Furthermore, the approach is

transparently described, and the used datasets are based on published ones. Making the datasets as used publicly available would be beneficial. The statistical analysis, as far as it is needed and done, is appropriate and seems to be valid.

The level of detail of the contribution is appropriate to enable reproduce the work to a certain extent, especially when public availability of used data is given.

There are no concerns about inappropriate or even libelous language, diversity, equity and inclusion.

Suggested improvements

- A comparison of the application of the introduced solution with classical, i.e. non-LLM-based approaches, where available, would be beneficial, especially when its quantified, and could be suggested for future work.
- Introducing all abbreviations, including but not limited to CNC, DNN, and CLIP, at their first use could be beneficial for readers.

Version 1:

Reviewer comments:

Reviewer #1

(Remarks to the Author)

Dear authors,

thank you for the revised manuscript, I have no more comments.

Reviewer #3

(Remarks to the Author)

Comment	Rebuttal
Reviewer 1	
Thank you for this interesting manuscript for Communications Engineering. The current advance of LLM use in engineering disciplines is manifold and your contribution of a chat-bot for CAD/CAE-based engineering data retrieval and analysis is valuable. There are some potentials to improve the manuscript which I briefly outline below:	Thanks for the positive feedback! Modifications regarding the constructive suggestions have been carefully made.
- Framing: In the beginning of your manuscript, you make some bold statements such as "create a conversational interface that can respond with a holistic knowledge of all aspects of mechanical engineering". I would suggest to make it a size smaller. Although I understand the long-term objective, the system you describe is from my understanding far away from this. Perhaps it would be good to state it a bit more neutral as I did above in my comment on the chat-bot.	The authors have responded to the reviewers comment by rewording the aim to be "Motivated by this vision the aim of the work reported here is to create a prompt driven chatbot that can reason and analyse CAE data relating to components made available to it." 6th paragraph of Introduction section.
- Literature work in the introduction: Please also use sources for the main statements in your introduction.	More references (1-6) have been in the Introduction section.
- CAE data analysis: I wonder why you limit yourself on sub-symbolic AI approaches. There are plenty knowledge-based engineering systems using symbolic AI that can support or complement your approach and should be at least mentioned in a holistic view on this research field.	Thanks for this valuable suggestion. We have not included text and references to literature on sub-symbolic AI at the end of the subsection 'CAE Data Analysis' in the Related work section.
- Method description: Please correct the number of use cases. In the body text you mention four, in the list you mention only three.	The text has been corrected.
- Level 1 to 3 tasks: You state queries here. I understand what you mean by this but it would be favourable if you could abstract this to a higher level. E.g. Level one targets at data that is accessible either by CAD-part properties or PDM system outputs for a distinct part number without computation,	The authors agree that the reviewer's suggestion would enhance the paper and have improved the descriptions of the task levels in the 2nd -4th paragraphs in the subsection 'CAE data sources' in the Methods section.

level 2 involves PDM for multiple part numbers, Level 3 addresses other data structures or, as you state correctly for the experiments part, the answering capabilities of the LLM backbone.	
- Data sources: Please explain a bit more in detail what you understand as implicit knowledge. From my point of view, everything you involve is explicit knowledge which can be retrieved e.g. from work plans, PLM systems, etc. You are right when you restrict this simply to the CAD-model.	The authors agree that the current explanation of implicit and explicit is not clear. The 1st paragraph of the subsection entitled ‘CAE data sources’ in Methods section has been rewritten to better define the phrases ‘explicit’ and ‘implicit’ data with respect to digital representations.
- Embedding method: In some other papers, the authors made good experience with BLIP2 as data extraction method for multi-modal data. Something that accounts for all your choices: Please explain why you chose the methods/algorithms and what would be alternatives?	The embedding models used in this work are applied for text-image modality alignment. CLIP was selected due to its established performance, widespread adoption, and strong benchmarks for image-text contrastive learning. ViLT was chosen as a comparative model to CLIP because, unlike CLIP or BLIP, it does not rely on a convolutional or vision backbone; instead, it uses a transformer-based architecture throughout, making it a complementary and computationally lightweight alternative. While we acknowledge the reviewer’s suggestion that BLIP is a strong candidate, particularly given its success in prior work, BLIP is primarily optimized for generative vision-language tasks (e.g., captioning and VQA) and is more computationally intensive. In contrast, our application emphasizes efficient modality alignment rather than generation, and therefore CLIP and ViLT were selected for their lightweight architectures and alignment-focused design.
- Level 3 Experiments: I'm not convinced of your experiments. Just relying on the examples you state as dialogue, I would not agree that the system is able to give you any information. From my understanding, the question which part is the most expensive in manufacturing there is plenty of information missing to judge this and which are highly context sensitive (lot size, necessity for jigs, process chains involved, precision and	The reviewer questions the level 3 experiments which prompt the LLM to identify the ‘most expensive component’ and, separately, asks for the most similar component based on the modelling history. The level 3 experiments require the LLM to combine the knowledge of specific components (that has been made available to the system via RAG) and its “general knowledge” (from its pre-training). The

tolerances, achievable surface tolerances, ...) which the system is incapable to either retrieve or interpret without further instructions. The same accounts for the history generation. Without a clear purpose of the part, the modeling history is basically interesting but you need to get to a part geometry anyways. So please reflect on the relevance of the level 3 experiments.	aim of the level 3 prompts is not to generate perfect but rather demonstrate that the LLM is able to mix the specific and general knowledge, and reason about its responses with the limited given information in comparison to human engineers. The reviewer's remark indicates that this was not clearly explained and so the text has been edited in the section entitled "Experiments for Level 3 tasks" to better explain the significance of the results. Specifically, the 1st and 2nd paragraphs have been rewritten to make the objectives of the level 3 experiments clear.
Future work: This sub-section is a bit flat. Although I fully understand the direction, I would recommend to make it a bit more substantial and name concrete steps either for integrating additional data sources (FEM would be interesting but is a completely other data structure and already poses some challenges in CAD-FEA-integration) or in enhancing the result quality of the LLM-based agents.	Thanks for this constructive suggestion. The 'Future Work' section has been modified to add 1) incorporating more engineering data sources such as MRP data into the framework, 2) benchmarking MechRAG's performances on traditional engineering problems for the future work.
Reviewer 2	
This paper introduces MechRAG (Mechanical Retrieval-Augmented Generation), a multimodal large language model architecture designed to unify information from multiple engineering representations typically found in Computer-Aided Engineering (CAE) and Computer-Aided Design (CAD) environments. The authors posit that MechRAG attains a high degree of accuracy in the execution of mechanical activities that are commonly performed, including data management and classification tasks. They further contend that MechRAG effectively replicates engineer-level reasoning in contexts that are more inferential and subjective. It is suggested that such conversational interfaces have the capacity to enhance engineering productivity to a significant degree, facilitate new interaction paradigms, and drive transformative workflows across a variety of stages in the design and	Thanks for the positive feedback! Modifications regarding the constructive suggestions have been carefully made.

manufacturing process.

MechRAG has the capacity to amalgamate a variety of informational sources, encompassing both visual and textual elements.

The paper sets out a three-tiered structure for tasks, with each tier exhibiting a distinct level of demand. The levels are designated as direct, collective and emergent. The integration of diverse informational sources is facilitated by the utilisation of embedding methods, thereby enabling the execution of varied experiments at different levels. A total of five different configurations of variables are tested for MechRAG, as illustrated in Figure 3. MechRAG's capabilities and potential are evaluated through a series of tasks that span levels 1 to 3. The assessment encompasses a diverse range of question types and levels of difficulty, thereby ensuring a comprehensive evaluation of the candidate's aptitude and proficiency.

In the context of multi-modal classification tasks and information queries concerning individual models, MechRAG demonstrated a noteworthy level of proficiency.

The accuracy levels achieved were 89.41% and 90.48%, respectively. The following text is intended to assist with the answering of questions and the completion of inferences tasks.

In relation to the aggregation of models and the entire dataset, human engineers have reported that MechRAG has achieved a level of proficiency comparable to that of human operators.

The following section will address the topic of performances. A salient constraint of the present implementation of MechRAG pertains to the limited amount of data that can be utilised to augment LLMs.

The work presented is of interest, and the various experiments conducted with different variable configurations are also of interest. The authors employed the system to address repetitive and classification tasks at varying levels of complexity. It is evident that this instrument has the capacity to enhance the productivity of engineers,

constituting a significant research endeavour.	
Finally, what will be the possible improvements that the authors can imagine for MechRAG in addition of the token limits to improve the performance levels?	The ‘Future Work’ section has been modified to add 1) incorporating more engineering data sources such as MRP data into the framework, 2) benchmarking MechRAG’s performances on traditional engineering problems for the future work.
Reviewer 3	
A multimodal LLM for Mechanical Engineering” by Shuang Li and Jonathan Corney The authors introduce a multi-modal large language model (LLM) architecture, Mechanical Retrieval-Augmented Generation (MechRAG). For this, the contribution starts with a motivation, based on this a vision and objectives. Related work is highlighted for the areas of LLMs, Retrieval-Augmented Generation (RAG)-systems, and Computer-Aided Engineering (CAE) data analysis. On this foundation, the method of the work is introduced, including three levels of tasks associated to engineering activities and data, which are addressed by the developed solution, as well as related data sources and datasets. The three stages of MechRAG, indexing, retrieval, and generation, are highlighted and five configuration variables for MechRAG-setups and related experiments are described. The variables are embedding method, embedding model, indexing ratio, prompting strategy, and LLM platform. Then, experiments for the three defined levels are executed and the related results discussed. The contribution ends with a conclusion, discussion and outlook to future work. Abstract, introduction, and conclusion are clear and appropriate. References contain essential pre-existing work as far as known. Key result and major claim of the paper is the development and evaluation of a multi-modal LLM architecture for engineering data, and related analysis tasks, respectively. This is a novel approach with a high degree of originality and may be of high interest to the community and potentially beyond.	Thanks for the positive feedback! Modifications regarding the constructive suggestions have been carefully made.

The contribution is well written and convincing; the chain of reasoning is good to follow and adequately presented. The contribution has the potential to influence the thinking in the area of software support for mechanical engineering data analysis and creation. Validity of the approach is given based on its high degree of thoughtfulness and structure. Furthermore, the approach is transparently described, and the used datasets are based on published ones. Making the datasets as used publicly available would be beneficial. The statistical analysis, as far as it is needed and done, is appropriate and seems to be valid. The level of detail of the contribution is appropriate to enable reproduce the work to a certain extent, especially when public availability of used data is given. There are no concerns about inappropriate or even libelous language, diversity, equity and inclusion.	
Suggested improvements  • A comparison of the application of the introduced solution with classical, i.e. non-LLM-based approaches, where available, would be beneficial, especially when its quantified, and could be suggested for future work. 	Thanks for this valuable suggestion. The ‘Future Work’ section has been modified to add benchmarking MechRAG’s performances on traditional engineering problems for the future work.
 • Introducing all abbreviations, including but not limited to CNC, DNN, and CLIP, at their first use could be beneficial for readers. 	All abbreviations have been checked and added explanation for the first time of use.

Comment	Rebuttal
Reviewer 1	
Dear authors, thank you for the revised manuscript, I have no more comments.	Thanks for the positive feedback!